# Efficiency of Tris-Based Extender Steridyl for Semen Cryopreservation in Stallions

**DOI:** 10.3390/ani10101801

**Published:** 2020-10-04

**Authors:** Elena Nikitkina, Artem Musidray, Anna Krutikova, Polina Anipchenko, Kirill Plemyashov, Gennadiy Shiryaev

**Affiliations:** Russian Research Institute for Farm Animal Genetics and Breeding—Branch of the L.K. Ernst Federal Science Center for Animal Husbandry, Moskovskoye sh. 55A, St. Petersburg, Pushkin 196625, Russia; 13linereg@mail.ru (A.M.); anntim2575@mail.ru (A.K.); aps.vet.93@yandex.ru (P.A.); kirill060674@mail.ru (K.P.); gs-2027@yandex.ru (G.S.)

**Keywords:** semen freezing, semen extender, stallions, semen quality, fertilizing ability, Tris

## Abstract

**Simple Summary:**

The cryopreservation and long-term storage of semen is one of the methods for accelerated improvement of the genetic qualities of animals. However, horse breeders prefer to use fresh or chilled semen, as the fertilizing capacity of frozen equine semen is much lower. It is important to find extenders, or a combination of extenders, that will improve semen survival after freezing. It is also important that the extender can be easily and simply prepared for use. Steridyl is a concentrate to which you just need to add sterilized water. This extender was developed for ruminants. In this study we tested Steridyl for freezing stallion semen. The motility, morphology, energy metabolism, DNA damage, and fertility of sperm frozen in Steridyl were evaluated. As a result, Steridyl was shown to be a good extender for equine semen freezing.

**Abstract:**

The fertilizing ability of stallion sperm after freezing is lower than in other species. The search for the optimal extender, combination of extenders, and the freezing protocol is relevant. The aim of this study was to compare lactose-chelate-citrate-yolk (LCCY) extender, usually used in Russia, and Steridyl^®^ (Minitube) for freezing sperm of stallions. Steridyl is a concentrated extender medium for freezing ruminant semen. It already contains sterilized egg yolk. Semen was collected from nine stallions, aged from 7 to 12 years old. The total and progressive motility of sperm frozen in Steridyl was significantly higher than in semen frozen in LCCY. The number of spermatozoa with normal morphology in samples frozen in LCCY was 60.4 ± 1.72%, and with Steridyl, 72.4 ± 2.10% (*p* < 0.01). Semen frozen in Steridyl showed good stimulation of respiration by 2.4-DNP, which indicates that oxidative phosphorylation was retained after freezing–thawing. No differences among the extenders were seen with the DNA integrity of spermatozoa. Six out of ten (60%) mares were pregnant after artificial insemination (AI) by LCCY frozen semen, and 9/12 (75%) by Steridyl frozen semen. No differences among extenders were seen in pregnancy rate. In conclusion, Steridyl was proven to be a good diluent for freezing stallion semen, even though it was developed for ruminants.

## 1. Introduction

Artificial insemination is one of the main methods of genetically improving the horse population, due to the accelerated use of the best stallions. Frozen semen has a number of advantages over fresh or chilled semen. Frozen semen preserves the horse’s genetics for a long time, even after death, allowing the material to be transported over long distances, around the world. In modern assisted reproductive technology programs such as in vitro fertilization (IVF) and intracytoplasmic sperm injection (ICSI), cryopreserved sperm are mainly used. Artificial insemination of horses is widespread. However, breeders prefer to use fresh or chilled semen [1,2]. The fertilizing ability of stallion sperm after freezing is lower than in other species [3,4]. One of the reasons is the selection of stallions based on the exterior sports quality, and productivity, and not focused on reproductive qualities [4,5]. Sperm freezing in horses is very important for the preservation of valuable, rare, and endangered genotypes [6,7,8].

The search for the optimal extender, combination of extenders, and the freezing protocol is relevant. Semen extenders affect semen quality, especially after freezing [6,9]. Extenders usually are composed of sugars, electrolytes, egg yolk, and skim milk [10]. Skim milk-based extenders are effective in freezing stallion semen. Lipoproteins in skim milk protect spermatozoa from cold shock [10]. Chicken yolk is also one of the main components of the medium. The role of egg yolk in the cryopreservation of sperm is to be a resistance factor, which helps to protect against cold shock, and a storage factor, which helps to maintain viability [5]. The phospholipid, cholesterol, and the low-density lipoprotein content of chicken egg yolk have been identified as the protective components [11]. Most commercial extenders require the addition of fresh egg yolk. However, milk and egg yolk are biological fluids, and may contain components that are unfavorable for stallion semen. For example, α-lactoglobulin has been shown to be deleterious to the survival of equine sperm [12]. It has been reported that the source of egg yolk in Tris-based extenders affects semen motility in chilled dromedary camel ejaculates [13], and on post-thawing sperm motility and fertility in buffalo [14]. This could be attributed to the differences in fatty acid, phospholipids, cholesterol, and lipoprotein levels in egg yolk [15].

Steridyl^®^ (Minitube, Germany) is concentrated extender medium for freezing of bull semen, and semen of other ruminants. Steridyl is a Tris based extender. The major benefit of Steridyl is that it already contains sterilized egg yolk in the concentrate. This eliminates the time-consuming preparation of fresh egg yolk. In this study we tried to freeze equine semen, diluted with Steridyl. We chose lactose-chelate-citrate-yolk (LCCY) extender for control, as it is used in Russia in most cases, and is recommended by the requirements of the Russian Federation.

The aim of the study was to compare LCCY extender, usually used in Russia, and Steridyl for freezing sperm in stallions.

## 2. Materials and Methods

### 2.1. Ethics Statement

The principles of laboratory animal care were followed, and all procedures were conducted according to the ethical guidelines of the L.K. Ernst Federal Science Center for Animal Husbandry and the Law of the Russia Federation on Veterinary Medicine No. 4979-1 (14 May 1993).

### 2.2. Animals

Nine stallions aged from 7 to 12 years old were used for this study. Semen was routinely collected 1–2 times a week during breeding season (March–June 2019).

### 2.3. Chemicals and Extenders Preparation

The chemicals used for the preparation of lactose-chelate-citrate-yolk (LCCY) extender and centrifugation solution were ordered from Sigma-Aldrich (Sigma, Saint Louis, MO, USA). Egg yolk was collected from chickens of the bioresource collection “Genetic Collection of Rare and Endangered Chicken Breeds” (RRIFAGB, Pushkin, St. Petersburg, Russia). Steridyl contains Tris, citric acid, sugar, buffers, glycerol, purified water, irradiated sterile egg yolk, and antibiotics (Tylosin, Gentamicin, Spectinomycin, Lincomycin). The chemicals proportion is the company’s trade secret. Lactose-chelate-citrate-yolk (LCCY) extender consisted of 321 mM lactose, 3 mM ethylenediaminetetraacetic acid dinatrium salt (Trilon B), 3 mM sodium citrate, 0.95 mM sodium bicarbonate, 20% freshly prepared egg yolk, and 0.4 mg/mL gentamicin. Steridyl was prepared through a 1:1.5 dilution with sterilized water. The centrifugation solution consisted of 204 mM lactose, 25 mM glucose, 3 mM ethylenediaminetetraacetic acid dinatrium salt (Trilon B), 0.4 mM magnesium sulfate, 21 mM sodium chloride, and 14 mM potassium citrate.

### 2.4. Semen Collection and Preparation

The semen was collected with a Hannover artificial vagina (Minitüb GmbH, Tiefenbach, Germany). A total of 39 ejaculates (3–4 ejaculates per stallion) were collected. Immediately after collection, the semen was divided into two equal parts. One part (P1) was diluted by lactose-chelate-citrate-yolk (LCCY) extender, and a second part (P2) was diluted by centrifugation extender. Dilution ratio was 1:1. Sperm concentration, total (TM) and progressive motility (PM) was evaluated by a computer-assisted sperm analysis (CASA) in a Mackler chamber at 37 °C. The Argus CASA system (ArgusSoft LTD., St. Petersburg, Russia) and a Motic BA 410 microscope (Motic, Hong Kong, China) were used. Ejaculates with volume less than 30mL, concentration less than 100 × 106 sperm/mL, and total motility less than 60% were excluded from the study. A total of 8 ejaculates were excluded from the study. A total of 31 ejaculates were chosen for cryopreservation. 

### 2.5. Cryopreservation

The diluted samples were centrifuged for 8 min at 600 g, the supernatant was eliminated and the P1 was resuspended in LCCY medium containing 3.5% glycerol, and the P2 was resuspended in Steridyl^®^ medium (Minitüb GmbH, Tiefenbach, Germany). Final concentration was 200 × 10^6^ cells/mL. Semen was loaded into 0.5 mL straws and equilibrated at +5 °C for 120 min. The straws were frozen in liquid nitrogen vapor at −110 °C for 12 min, and stored in a liquid nitrogen tank. 

### 2.6. Semen Evaluation after Thawing

After at least 24 h, semen was thawed at 37 °C for 30 s, and the contents of the straw were emptied into a 1.5 mL microcentrifuge tube. Semen was evaluated for total and progressive motility as previously described.

Morphology was assessed using Diff-Quick kit (Abris+, St. Petersburg, Russia) stained smears and CASA (ArgusSoft–module Morphology). The morphology of at least 250 spermatozoa was examined on each slide at 1000× magnification with immersion oil. 

We used a sperm chromatin dispersion (SCD) test [16] to assay DNA fragmentation in semen. GoldCyto DNA kit (Guangzhou, China) was used for the SCD test. A total of 100 spermatozoa per sample were measured by CASA (ArgusSoft–module DNA fragmentation).

Oxidative phosphorylation (OXPHOS) was assessed by the reaction of cellular respiration rate (CR) on adding 2,4-dinitrophenol (2.4-DNP) to the semen sample [17]. The addition of 2.4-DNP disrupts the proton gradient by carrying protons across a membrane, and uncouples proton pumping from ATP synthesis, because it carries protons across the inner mitochondrial membrane. As a result it increases respiration rate. If the respiration rate increases, there is good OXPHOS. We evaluated the CR using an ion meter “Expert-001MTX” and Clarke electrode (Research and Production Company “Econix-Expert”, Moscow, Russia). Next, 100 μL of semen was added to the chamber with 1 mL of 11% lactose, and the rate of decrease in oxygen concentration was measured. Then, 10 μL 2.4-DNP was added. The ratio of the respiration rate with 2.4-DNP to the respiration rate before adding 2.4-DNP was found. Oxidative phosphorylation can be considered good when the rate of cellular respiration increases two or more times after the addition of 2.4-DNP.

### 2.7. Artificial Insemination (AI) and Pregnancy Control

Ten mares were inseminated with LCCY frozen sperm and twelve mares were inseminated with Steridyl frozen sperm. The mares were in good health. Artificial insemination was carried out in accordance with the breeding program. Doses frozen in Steridyl or in LCCY from the selected stallions were used randomly. AIs were performed by transcervical injection of the dose of thawed sperm, with preliminary control of follicle development 0–6 h after ovulation. The dose of frozen sperm was 250 million spermatozoa with PM. Pregnancy was determined by ultrasound examination on the 14th day after ovulation. The number of pregnant mares was counted in each AI cycle.

### 2.8. Statistical Analysis

For statistical analysis, the computer software IBM-SPSS Statistics 19 (IBM, Armonk, NY, USA) and Statistica 10 (TIBCO Software Inc., Palo Alto, United States) were used. All data were normally distributed (Kolmogorov–Smirnov test). Data were analyzed by ANOVA. The data were expressed as means ± standard error of the mean. All pairwise multiple comparisons between means were conducted by *t*-test. We compared pregnancy results by z-test for two proportions. Differences were considered statistically significant at *p* < 0.05.

## 3. Results

Semen used in this study was in the acceptable range for sperm motility and morphology after dilution. Total motility, progressive motility, and spermatozoa of normal morphology of semen diluted in Steridyl or LCCY extenders are shown in Table 1. No differences between the two extenders were found. 

Characteristics of semen quality frozen in Steridyl or LCCY extenders are shown in Table 2. The post thaw total and progressive motility of semen frozen with Steridyl were higher than when frozen with LCCY (*p* < 0.05). Six stallions classified as ‘good freezers’ had 41.2 ± 1.13% progressive motility in samples frozen with Steridyl, and 37.2 ± 1.02% progressive motility in samples frozen with LCCY. Three stallions classified as “poor freezers” had 29.7 ± 2.93% progressive motility in samples frozen with Steridyl, and 18.0 ± 0.73% progressive motility in samples frozen with LCCY. The number of spermatozoa with normal morphology in samples frozen in LCCY was lower than with Steridyl (*p* < 0.01). No differences among extenders were seen with the DNA integrity of spermatozoa.

Pregnancy rates (pregnancy confirmation) at day 14 are shown in Table 2. A total of 6/10 (60%) mares were pregnant after artificial insemination (AI) by LCCY frozen semen, and 9/12 (75%) by Steridyl frozen semen. No statistical difference was found between groups.

The increase of cellular respiration rate after the addition of 2.4-DNP was 1.92 ± 0.04 times in LCCY frozen sperm, and 2.21 ± 0.07 times in Steridyl frozen sperm (Figure 1). Oxidative phosphorylation was better in Steridyl than in LCCY (*p* = 0.0005).

## 4. Discussion

There are many factors that affect the quality of frozen semen in stallions. These are freezing protocols, extender formulation, and type of cryoprotectant. The composition of extender is one of the main factors in determining the viability of frozen semen [18]. Although most extenders for freezing equine semen provide sperm viability and pregnancy rates, new freezing extenders or modifications of freezing protocols are still required [4,18,19,20]. Stallion semen freezing protocols vary from laboratory to laboratory, using different extenders, both commercial and homemade, and different cryoprotectants. Often the protocols and extenders are selected individually for the stallion. It is reported that only 30–40% of stallions produce sperm with good cryoresistance [5,21]. Finding the extender that is easy to prepare and suitable for most stallions is important.

Egg yolk is a common component of most semen extenders for domestic animals. It has been shown to have a positive effect on sperm, and to protect acrosomes and the plasma membranes against cold shock [22]. When using commercial diluents, you have to add the egg yolk prior to use. The chicken yolk must be added before using most extenders. It is not possible to use yolks of the same quality, so the quality of the extender may vary. In this study, we used the commercial extender Steridyl to freeze semen from stallions. Steridyl was developed to freeze ruminant semen. Its main advantage is that it already contains sterile egg yolk, and you just need to add sterile water to the concentrate [23]. 

The results of our study showed that total and progressive motility of frozen semen, on average, met the requirements for sperm use for AI [2,24]. It is generally accepted that, on average, even under ideal conditions, 40–50% of spermatozoa do not survive freezing [25]. Vidament suggested that 35% post thawing motility is sufficient for insemination of mares [20]. Loomis and Graham suggested that 30% is enough. However, many stallions do not have even 30% motility after thawing [2]. Stallions with less than 20% motility after freezing may be qualified as “poor freezers’”. The total and progressive motility of sperm frozen in Steridyl was significantly higher than in semen frozen in LCCY. Stallions classified as “good freezers” had higher motility in both extenders than stallions classified as “poor freezers”. One “poor freezer” had less than 20% progressive motility in both extenders. Two “poor freezers” had less than 20% progressive motility in LCCY frozen samples, and 35% or more in Steridyl frozen samples. So Steridyl can be used in “poor freezers”. 

Usually, the assessment of motility is the first and main parameter by which to judge the survival of sperm after freezing [26]. However, deeper tests are also needed. Sometimes sperm with high abnormal morphology has good motility [27]. In our studies, sperm frozen in Steridyl had better morphology than in LCCY. Although some authors have noted that the extenders do not affect the morphology [26,28]. The main cell injuries that were encountered in our study were in the tail and neck of the spermatozoa. There were thickened necks, twisted and broken tails, swollen and wrinkled acrosomes, and missing acrosomes. Stallions differed in sperm morphology after cryopreservation. Some stallions had 55–58% spermatozoa of normal morphology in LCCY semen, and 60–65% in Steridyl frozen semen. Some stallions had more than 70% spermatozoa of normal morphology in both extenders. 

The intensity of energy metabolism is a very important criterion to assess sperm quality [29,30]. Oxidative phosphorylation (OXPHOS) is the metabolic pathway in which cells use enzymes to oxidize nutrients, thereby releasing energy which is used to produce adenosine triphosphate (ATP). Glycolysis predominates in bull and ram semen. But stallion spermatozoa is mainly dependent on the mitochondrial oxidative phosphorylation pathway to produce ATP [31,32]. Mitochondria are one of the cell structures most sensitive to the damaging effects of low temperatures [29,33]. Respiratory response to 2.4-DNP supplementation is a good OXPHOS test. The positive correlation between conception rate in cows and stimulation of the respiration by 2.4-DNP of bovine sperm after freezing was found (r = 0.62, *p* < 0.05) [34]. In our studies, semen frozen in Steridyl showed good stimulation of the respiration by 2.4-DNP, which indicates that oxidative phosphorylation was retained after freezing–thaw. We also observed individual variation between stallions and ejaculates in the response of cellular respiration to 2.4-DNP additions. In five samples, frozen both in Steridyl and in LCCY, respiration stimulation with 2,4 dinitrophenol was one, which means that there was no reaction to dinitrophenol, and cellular respiration was without oxidative phosphorylation. 

Many authors have suggested that DNA fragmentation affects fertility [8,35,36,37,38]. The sperm DNA was thought to be “dysfunctional” until fertilization took place. During the process of spermatogenesis, a high degree of compaction of sperm chromatin and associated nucleoproteins occurs. This degree of compaction is necessary to protect the DNA of the sperm during transport through the male and female reproductive tract, and for proper fertilization and embryo development [35]. Sperm DNA quality is associated with early embryonic death [39]. When evaluating sperm frozen in Steridyl and LCCY, DNA fragmentation was quite high. Our data are consistent with data of other authors. Sperm frozen in Steridyl and LCCY were not significantly different. 

Fertility is the main indicator of sperm quality. Fertility is influenced by many factors, one of which is the extender [4,6,18,26]. In our study, semen frozen in both diluents had good fertility. No statistical difference was found between groups, but pregnancy confirmation on day 14 was a little higher in Steridyl frozen sperm. A very small number of inseminations were performed, and this low number might be the reason why there were not differences in the pregnancy rate between extenders.

This study showed a high tolerance of equine semen for cryopreservation in Steridyl. The quality of the semen frozen in Steridyl was superior to the quality of the semen frozen in the LCCY in almost all studied parameters. Perhaps this is due to the quality of the yolk, or perhaps Steridyl has a better effect on the sperm ability to survive during freezing procedure. At the same time, individual variability in sperm quality after freezing was observed in different stallions. 

Steridyl is Tris-based extender. Tris-based extenders have been successfully used to freeze the semen of other animals [22,26]. However, previous work indicates poor equine sperm survival after freezing with Tris [26]. The authors point to the toxic effect of Tris-buffer due to its ability to penetrate the sperm membrane and alter intracellular metabolism. However, in our study, sperm metabolism was not severely impaired, and according to the reaction of the cellular respiration rate to the addition of 2.4-dinitrophenol, metabolism was better than in LCCY frozen semen. Sugar-lactose, glucose, fructose are also added to the diluent. Sugars are energy substrates, and they also protect sperm from osmotic shock and the formation of ice crystals inside and outside cells. Steridyl contains fructose and LCCY contains lactose. Pojprasath et al. noted in their study that fructose was less effective than glucose and sorbitol in protecting stallion sperm during cryopreservation [18]. In our study, the fructose containing diluent was better than lactose. The exact composition of Steridyl is not known, Steridyl’s composition and the ratio of its components may have better protective properties than other studied Tris-based equine extenders.

## 5. Conclusions

Steridyl has proven to be a good diluent for freezing stallion semen, although it was developed for ruminants. Steridyl can be recommended for semen freezing of both “good freezer” and “poor freezer” stallions.

## Figures and Tables

**Figure 1 animals-10-01801-f001:**
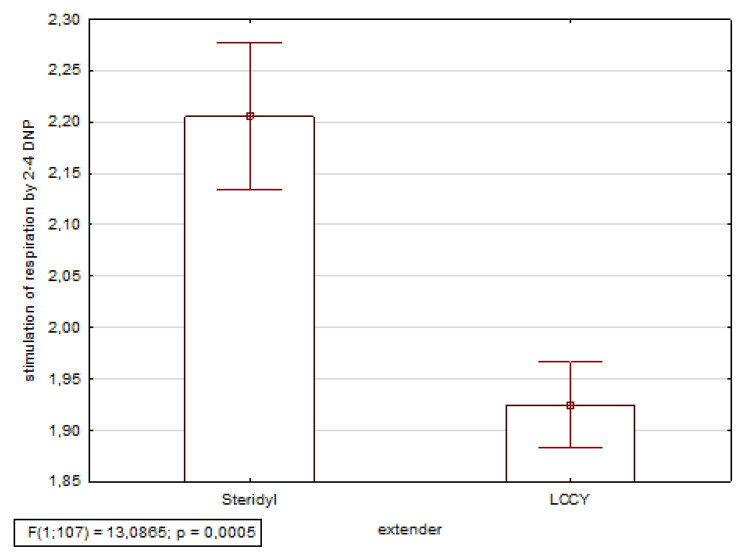
Stimulation of respiration by 2.4-dinitrophenol (2.4 DNP) in Steridyl frozen sperm and LCCY frozen sperm. The means ± SEM are presented.

**Table 1 animals-10-01801-t001:** Quality parameters (means ± SEM) in stallion semen samples (*n* = 31) diluted in Steridyl and lactose-chelate-citrate-yolk (LCCY) extenders.

	Steridyl	LCCY
Total Motility, %	83.2 ± 1.56	83.6 ± 1.97
Progressive Motility, %	75.3 ± 1.93	75.3 ± 1.38
Spermatozoa of Normal Morphology, %	75.8 ± 2.03	76.2 ± 1.82

*n*—number of samples for each extender.

**Table 2 animals-10-01801-t002:** Quality parameters (means ± SEM) in stallion semen samples (*n* = 31) frozen/thawed in Steridyl and LCCY extenders.

	Steridyl	LCCY
Total Motility, %	43.1 ± 1.86 ^a^	39.6 ± 0.93 ^b^
Progressive Motility, %	36.3 ± 2.14 ^a^	31.7 ± 1.13 ^b^
Spermatozoa of Normal Morphology, %	72.4 ± 2.10 ^c^	60.4 ± 1.72 ^d^
Spermatozoa with Damaged DNA, %	27.2 ± 6.16	29.3 ± 4.32
Pregnancy Rates (Pregnancy Confirmation) at Day 14, %	75 ± 12.5	60 ± 15.5

*n*—number of samples for each extender. ^a,b^ Differences are significant for *p* < 0.05. ^c,d^ Differences are significant for *p* < 0.01.

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
