# Peer review of "Efficiency of Tris-Based Extender Steridyl for Semen Cryopreservation in Stallions"

_animals, 2020, doi:10.3390/ani10101801_

Round 1

Reviewer 1 Report

In this manuscript, commercially available steridyl medium was used to increase the cryopreservation effect of stallions semen. And the results were compared with the diluent which is widely used for cryopreservation of horse sperm in Russia. The reviewer believes that there is no big problem with the experiment method or result. However, what the reviewer would like to point out is that the content of the thesis is a simple comparison of the effectiveness and lacks the originality that should be possessed as an academic paper. Also, at least the authors used steridyl medium as an experimental material, and it is the most important keyword in the paper, so I think authors should mention basic information about steridyl medium. Of course, it will be confidential to a specific company, but readers of the paper have the right to know what steridyl medium is. If there is no information about steridyl medium, the result that steridyl medium is effective for freezing sperm of horses is not differentiated from product advertisement brochures.

In order to make the thesis more clear and concise, I think it is better to omit Table 1(description in the result section) and to integrate Tables 2 and 3. And in Figure 1, need to insert a title on the x-axis.

Author Response

Point 1: I think authors should mention basic information about Steridyl medium.

Response 1: Basic information about Steridyl medium was added in the text. L-86-88 (Steridyl contains TRIS, citric acid, sugar, buffers, glycerol, purest water, irradiated sterile egg yolk and antibiotics (Tylosin, Gentamicin, Spectinomycin, Lincomycin). The chemicals proportion is the company's trade secret)

Point 2: In order to make the thesis more clear and concise, I think it is better to omit Table 1(description in the result section) and to integrate Tables 2 and 3. And in Figure 1, need to insert a title on the x-axis.

Response 2: Table 2 and Table 3 were combined in one table describing semen quality after cryopreservation. A title on the x-axis was inserted in Figure 1.

Reviewer 2 Report

In this study the Authors tested the application of Steridyl, a commercial extender medium developed for freezing a ruminants semen, for semen cryopreservation in stallion. This is a quite interesting study with applicability in reproductive biotechnology in stallion, where low fertilizing ability is recorded after freezing. However, it is flawed by insufficiency in writing and presentation skill.

Specific comments

L25 – redundant information,

L32-33 Was pregnancy different between methods?

In the Introduction there is lack of information concerning why lactose-chelate-citrate-yolk (LCCY) extender was used as a control for Steridyl.

Materials and methods

L 94 -Please explain why semen cryopreserved in Steridyl was previously diluted in centrifugation extender.

L 102 -Why supernatant was removed?  it means that semen was cryopreserved without seminal plasma? Please explain if seminal plasma has a deleterious effect onsemen quality in stallion?

L 104 - Please specify which Steridyl was used for cryopreservation of stallion semen.

L 106 – how was the temperature monitored?

L118-126 – please add reference for Oxidative phosphorylation measurement in semen.

Results

Table 2, Table 3 and Fig.1, should be combined in one table describing semen quality after cryopreservation

Discussion

Please do not repeat results in discussion.

L202: show characteristics of good and poor freezer in Results section

L:215-238, do not describe method in the discussion section.

L-250 move to M&M section

A lot of redundant information should be eliminated from the manuscript (e.g. L25-26, 49-50, 121)

Author Response

Point 1: L25 – redundant information

Response 1: L 25 – «The major benefit of Steridyl that it already contains the sterilized egg yolk in the concentrate». We chose this commercial extender mainly due to the fact that the composition already contains sterilized yolk. Therefore, we believe that this is important information. We have changed the sentence to « It already contains sterilized egg yolk»

Point 2: L32-33 Was pregnancy different between methods?

Response 2: Pregnancy did not differ between methods. We have added this information to the abstract – L-33.

Point 3:  In the Introduction there is lack of information concerning why lactose-chelate-citrate-yolk (LCCY) extender was used as a control for Steridyl.

Response 3: We chose this extender for control as it is used in Russia in most cases and is recommended by the requirements of the Russian Federation. We have added this information – L-68-69

 Point 4: L 94 -Please explain why semen cryopreserved in Steridyl was previously diluted in centrifugation extender.

Response 4: According to our observations, the egg yolk, which is part of Stediryl, interferes with centrifugation and precipitates with the cells. We used a special extender developed by us for centrifugation of stallion semen.  Combined use of centrifugation extender and Steridyl was  successful for cryopreservation of equine semen.

 Point 5: L 102 -Why supernatant was removed?  it means that semen was cryopreserved without seminal plasma? Please explain if seminal plasma has a deleterious effect onsemen quality in stallion?

 Response 5: Seminal plasma was removed and replaced with cryopreservation diluent. Removal of seminal plasma prior to cryopreservation was earlier shown to improve post-thaw sperm quality in stallions.

 Point 6: Please specify which Steridyl was used for cryopreservation of stallion semen.

 Response 6: We used Steridyl with antibiotics (ref: 13500/0260)

Point 7: L 106 – how was the temperature monitored?

 Response 7: We used a foam float that has a surface temperature of -110C when it floats in liquid nitrogen. When designing the float, the temperature was measured with a thermocouple

Point 8: L118-126 – please add reference for Oxidative phosphorylation measurement in semen.

Response 8: This method was developed at our institute. All publications are in Russian only. We have provided one of the most significant  reference – [17] - Moroz, L. G; Shapiev, I. Polarographic method for assessing energy metabolism in sperm by stimulating respiration with 2.4 - dinitrophenol. Bull. State Sci. Inst. All-Russian Res. Inst. Farm Anim. Genet. Breed. 1978, 33, 28–30.

Point 9: Table 2, Table 3 and Fig.1, should be combined in one table describing semen quality after cryopreservation

Response 9: Table 2 and Table 3 were combined in one table describing semen quality after cryopreservation. We believe that Figure 1 provides results of evaluation of sperm respiration by the addition of 2.4 DNP better than the table.

 Point 10: Please do not repeat results in discussion.

 Response 10: Thank you. We will take into account your remark

 Point 11: L202: show characteristics of good and poor freezer in Results section

 Response 11: We presented the characteristics of good and poor freezer in Results section (Six stallions classified as ‘good freezers’ had 41.2±1.13 % progressive motility in samples frozen with Steridyl and 37.2±1.02 % progressive motility in samples frozen with LCCY. Three stallions classified as ‘poor freezers’ had 29.7±2.93 % progressive motility in samples frozen with Steridyl and 18.0±0.73 % progressive motility in samples frozen with LCCY.) 

 Point 12:  L:215-238, do not describe method in the discussion section.

 Response 12: We have shortened the text. Part of the text was moved to the methodology section

 Point 13: L-250 move to M&M section

 Response 13:We moved L-250 to M&M section

 Point 14: A lot of redundant information should be eliminated from the manuscript (e.g. L25-26, 49-50, 121)

 Response 14: L25-26 - We chose this commercial extender mainly due to the fact that the composition already contains sterilized yolk. Therefore, we believe that this is important information. We have changed the sentence to « It already contains sterilezed egg yolk». L49-50 - Some information has been edited. L-121 - We consider this information important to understand the method.

Reviewer 3 Report

The study is of interest to standardize semen processing for cryopreservation in the stallion and support its implementation into practice.

The manuscript could be improved with some changes.

It is not clear if the effect of the breed was considered in the statistical analysis. Semen was collected from stallion of different breeds and, if replicates are present, the breed effect has to be considered in the statistical model.

Pregnancy was recorded after 14 days after ovulation in 22 mares and the frequency of pregnant mares was calculated as reported in table 3. Specify the statistics used to compare the two treatments providing no significant results. 

The style and titles of the tables need to be improved. I suggest to change rows into columns. Suggested title for Table 1: Quality parameters (LSMeans±SEM) in stallion semen samples (n=31) diluted in Steridyl and LCCY extenders. Suggested title for Table 2:  Quality parameters (LSMeans±SEM) in stallion semen samples (n=31) frozen/thawed in Steridyl and LCCY extenders.

Table 3 is redundant and results can be presented in the text.

English language needs to be improved and restyling of some sentences is recommended.

Author Response

Point 1: It is not clear if the effect of the breed was considered in the statistical analysis. Semen was collected from stallion of different breeds and, if replicates are present, the breed effect has to be considered in the statistical model.

 Response 1:   Our early studies showed that individual variability in sperm quality before and after freezing is higher than breed variability. In our experiment, there were only 9 animals. The mention of different breeds was removed from the text.

Point 2: Pregnancy was recorded after 14 days after ovulation in 22 mares and the frequency of pregnant mares was calculated as reported in table 3. Specify the statistics used to compare the two treatments providing no significant results.

Response 2: We compare pregnancy results by z-test for two proportions

Point 3: The style and titles of the tables need to be improved. I suggest to change rows into columns. Suggested title for Table 1: Quality parameters (LSMeans±SEM) in stallion semen samples (n=31) diluted in Steridyl and LCCY extenders. Suggested title for Table 2:  Quality parameters (LSMeans±SEM) in stallion semen samples (n=31) frozen/thawed in Steridyl and LCCY extenders.

Response 3: Thank you. We improved the style and titles of the tables.

 Point 4: Table 3 is redundant and results can be presented in the text.

 Response 4: Table 2 and Table 3 were combined in one table describing semen quality after cryopreservation

 Point 5: English language needs to be improved and restyling of some sentences is recommended.

 Response 5: Thank you. We will take into account your remark

Round 2

Reviewer 1 Report

I think everything the reviewer pointed out has been revised. The authors would like to make further revisions to improve the completeness of the manuscript and re-submit it.